# Detection of the Inoculated Fermentation Process of Apo Pickle Based on a Colorimetric Sensor Array Method

**DOI:** 10.3390/foods11223577

**Published:** 2022-11-10

**Authors:** Mengyao Wang, Jiawei Liu, Lu Huang, Haiying Liu

**Affiliations:** 1State Key Laboratory of Food Science and Technology, Jiangnan University, Wuxi 214000, China; 2School of Food Science and Technology, Jiangnan University, Wuxi 214000, China

**Keywords:** Apo pickle, colorimetric sensor array, fermentation, linear discriminant analysis, partial least squares regression

## Abstract

Apo pickle is a traditional Chinese fermented vegetable. However, the traditional fermentation process of Apo pickle is slow, easy to ruin, and cannot be judged with regard to time. To improve fermentation, LP-165 (*L. Plantarum*), which has a high salt tolerance, acidification, and growth capacity, was chosen as the starter culture. Meanwhile, a colorimetric sensor array (CSA) sensitive to pickle volatile compounds was developed to differentiate Apo pickles at varying degrees of fermentation. The color components were extracted from each dye in the color change profiles and were analyzed using principal component analysis (PCA) and linear discriminant analysis (LDA). The fermentation process of the Apo pickle was classified into four phases by LDA. The accuracy of backward substitution verification was 99% and the accuracy of cross validation was 92.7%. Furthermore, the partial least squares regression (PLSR) showed that data from the CSA were correlated with pH total acid, lactic acid, and volatile acids of the Apo pickle. These results illustrate that the CSA reacts quickly to inoculated Apo pickle and could be used to detect fermentation.

## 1. Introduction

Apo pickle is a traditional Chinese fermented vegetable that has been consumed in Suzhou and the surrounding areas for thousands of years. Traditionally, it is prepared with fresh Chinese flowering cabbage (*Brassica Campestris L.ssp.Chinensis var.utilis Tsen et Lee*) as raw material and is naturally fermented after salting [1]. Unlike the Sichuan pickle, which is fermented in brine at a low salt concentration (20–100 g/L) by the microorganisms on raw cabbage [2], the Apo pickle is fermented by directly mixing vegetables with salt. Apo pickle is often served as a side dish in traditional Chinese cuisine. Because of its unique appetizing features, which include sourness, aroma, umami, and crispness, it is popular with local people. However, because it is mostly homemade, the process is time-consuming and easy to ruin. Compared to traditional fermentation, adding bacterial strains to the fermentation ensures the quality of Apo pickle fermentation and allows it to be done quickly and efficiently.

Pickles are usually fermented spontaneously by lactic acid bacteria (LAB), acetic acid bacteria, yeast, and other environmental microorganisms [3]. The fermentation process can be divided into hetero-fermentation and homo-fermentation. The former is dominated by *Leuconostoc mesenteroides*, which produce distinctive flavors; the latter is dominated by *Lactobacillus plantarum,* which produce acid rapidly [4,5,6,7]. According to the Apo pickle production method, the starter cultures of Apo pickle fermentation are supposed to have high salt tolerance and acidification capacity.

The fermentation degree of pickles can be intuitively gained through sensory evaluation, but this is subjective, and participants require special training; in addition, the factory will also analyze the pickle products for biochemical indicators, such as volatile organic compounds (VOCs), acidity, nitrite, and so on, which will necessitate specialized equipment and complex processing [8,9,10]. These methods must also take the sample out of the fermenter, which raises the possibility of sample contamination. Therefore, more simple and less intrusive detection methods are necessary for detecting the degree of fermentation [11].

The colorimetric sensor array (CSA) has attracted increasing attention as a method for the classification and adulteration detection in chemically diverse analytes [12,13,14,15,16,17]. The CSA simulates the human olfactory system, presenting different combined responses to different compounds based on cross-reactive sensing receptors [18,19]. The advantage of this method is that it is real time, has a fast response time, does not require pre-treatment of the sample, and can effectively determine the condition of food. At present, CSA has been successfully applied to tea [20], vinegar [21], and *Tremella aurantialba* [22]. Based on the characteristics of CSA, it could also be used to detect and assess the fermentation process of the Apo pickle.

The aim of this study is to screen a strain to improve the fermentation process of the Apo pickle and prepare a colorimetric sensor array (CSA) to distinguish the progress of the fermentation process using a visual and non-destructive method.

## 2. Materials and Methods

### 2.1. Materials

Fresh Chinese flowering cabbage was purchased from a local supermarket (Wuxi, China). Indicators (bromocresol green, methyl red sodium salt, bromophenol blue, chlorophenol red, bromothymol blue sodium, and bromocresol purple), sodium hydroxide (NaOH), ethanol, phenolphthalein, sodium tetraborate decahydrate, potassium hexacyanoferrate (II) trihydrate, zinc acetate dihydrate, silver nitrate solution, and qualitative filter paper were bought from Sinopharm Chemical Reagent Co., Ltd. (Shanghai, China). *N*–(1–Naphthyl) ethylenediamine dihydrochloride, t-butylammonium hydroxide (TBAH), and p-toluene sulfonic acid (TsOH) were obtained from Macklin Biochemical Co., Ltd. (Shanghai, China). 5,5–dithiobis (2–nitrobenzoic acid; DTNB) was bought from Aladdin Biochemical Technology Co., Ltd. (Shanghai, China), and polyethylene glycol (PEG–400) was bought from Beijing InnoChem Science and Technology Co., Ltd. (Beijing, China). Man, Rogosa, and Sharpe (MRS) broth was purchased from Haibo Biotechnology Co., Ltd. (Qingdao, Shandong, China). Polyvinylidene fluoride (PVDF) membranes were purchased from Biosharp Biotechnology Co., Ltd. (Shanghai, China). All chemical reagents were of analytical grade and used as received.

### 2.2. Screening of Starter Cultures

Seven bacterial strains stored at low temperature in the laboratory were used in the study (Table 1). The seven strains were streak cultured on MRS agar for 48 h at 37 °C to obtain activated pure colonies. The pure colonies were then inoculated in 10 mL MRS broth and cultured for 24 h at 37 °C. The bacterial solution was harvested after centrifugation (8000 RPM, 10 min) and washed twice with sterile 0.85% saline; a concentration of 7 Lg CFU/g was prepared for use [23].

The MRS broth with 0%, 2%, 4%, 6%, and 8% salt concentrations were prepared. Each bacterial solution was added to MRS broth with different salt concentrations at 3% (*v/v*) inoculum. The optical density (OD) value and acidity were measured after 48 h of culture at 37 °C to test the salt tolerance of different strains.

According to a modified method of Choi et al., mixed vegetable/salt/water at a ratio of 50:4:46 and juice [24] were used, the filtrate was collected, and 3% (*v/v*) bacterial solution was added. The liquid was placed in an incubator at 25 °C, analyzed every 12 h for acidity and microbial number, and cultured for three days to test the adaptability of different strains on fresh Chinese flowering cabbage.

### 2.3. Apo Pickle Sample Preparation

Fresh Chinese flowering cabbage was washed, air dried, and mixed with 8% (salt/vegetable, *w/w*) concentration salt. Then 3% (fermentation strains/vegetable, *v/m*) fermentation strain was added to the mixture while placing a heavy weight on top. After 2 days, salt brine was dumped, and the jar was filled, sealed, and fermented at 25 °C. Pickle samples were collected daily for 0, 1, 2, 3, 5, 7, 11, and 15 days. Each sample consisted of three randomly selected jars for error reduction.

### 2.4. Biochemistry of Apo Pickle

A series of physicochemical examinations was undertaken according to the Chinese Industry Standard SB/T 10213–1994. The 10 g sample was placed in a 250 mL conical flask, then 50 mL of distilled water was added, and the sample was boiled in a water bath for 30 min. The filtrate was filtered through filter paper, and the volume of the collected filtrate was calibrated to 200 mL and mixed for the next assay. The pH value was measured directly with a pH meter, and the total acid was titrated with 0.01 mol/L NaOH. Salinity was calculated by direct titration with 0.1 mol/L silver nitrate solution. Nitrite was detected according to Chinese National Standards GB 5009.33–2016, a spectrophotometric method.

A 10 g pickle sample and 90 mL of 0.80% NaCl solution were mixed and homogenized using a blender, then diluted at a radio of 1:10. Subsequently, 0.1 mL of the mixture was evenly plated onto MRS broth [2]. All plate counts were repeated three times and incubated for 48 h at 37 °C.

### 2.5. Organic Acid Detection of Apo Pickle

A 2.0 g sample was placed in a 50 mL centrifuge tube, the volume was fixed to 10 mL, mixed, then transferred into an ultrasonic cleaning machine for extraction. After 30 min, the sample was removed and centrifuged (8000 RPM, 15 min), and the supernatant was filtered through a 0.45 μm water membrane.

The chromatographic column was a Zorbax Eclipse XDB C18 (Agilent Technologies Inc., Palo Alto, CA, USA), the column temperature 25 °C, and the mobile phase was a mixture of methanol and potassium dihydrogen phosphate solution (0.01 mol/L, pH 2.8) with a volume ratio of 5:95; the flow rate was 1 mL/min, and the injection volume was 5 µL. Isometric elution was performed. The detection wavelength of the UV detector was 210 nm.

### 2.6. Volatile Organic Compounds Detection of Apo Pickle

Approximately 5 g of chopped pickle sample was placed in a 20 mL headspace vial. Headspace solid-phase microextraction–gas chromatography–mass spectrometry was used to detect the VOCs of pickle samples.

The GC parameters were as follows: chromatographic column, Rix–17Sil MS (30 m × 0.25 mm × 0.25 μm); injector temperature, 250 °C; and carrier gas (He) velocity, 1 mL/min. The column temperature was kept at 40 °C for 3 min, increased to 230 °C at 10 °C/min, and then kept thus for 6 min.

The MS conditions were as follows: ion source, LECO EI; ion source temperature, 210 °C; emission current, 1 mA; electron energy, 70 eV; and transfer-line temperature, 250 °C. The mass scan range (m/z) was set from 33 to 400 amu in full-scan mode. The parameters used for identification and calculation of VOCs followed the protocol of Liang et al. [8].

### 2.7. Fabrication of the CSA

Normally, in a CSA, several indicators with different acid–base properties are used to cover a wide pH range. Firstly, 2 mg of the indicator was dissolved in 1 mL of ethanol to prepare indicator solutions. The pH values of gas-sensitive material solutions were adjusted with the addition of 0.8 mol/L TBAH or 1 mol/L Ts OH in water [25]. The amounts of 3 μL, 4 μL TsOH, and 0.5 μL, 1 μL, 6 μL TBAH were added to 500 μL Bromocresol green, Bromophenol blue, Methyl Red sodium salt, Bromothymol blue sodium, and Chlorphenol red solution, respectively, and mixed thoroughly to form the pH-adjusted dye solutions. DTNB was dissolved in phosphate buffer (pH 7.0) to prepare 1 mg/mL DTNB solution. PEG-400 was dissolved in ethanol to prepare a 0.1 g/mL PEG solution. The last three sulfur-derived dye solutions were prepared by mixing the solution of pH indicators, DTNB, PEG-400, and 0.1 mol/L NaOH according to the ratio 1:3:3:1 [26].

Nine gas-sensitive materials were obtained (Table 2). Two microliters of each indicator were pipetted and printed on the PVDF membrane by microcapillary pipettes to construct a 3 × 3 sensor array. After 30 min drying at 50 °C, the CSA was obtained and sealed at low temperature and protected from light place for storage.

### 2.8. CSA Data Acquisition

Images of the CSA were scanned by an HP LaserJet M1005MFP flatbed scanner (Hewlett Packard Inc., Shanghai, China) with a set resolution of 600 dpi. The “initial image” of the sensor array was first captured by the flatbed scanner before exposure to the pickle samples. The prepared CSA was quickly placed on the partition board of the reaction container. After 30 min of reaction, it was removed and scanned, and the scanned picture was named “final image” (Figure 1). In order to minimize data error caused by color inhomogeneity, the RGB (red, green, blue) values of 100 pixels near each selection point were averaged. The absolute value of the RGB difference (shown in equations below) of each indicator before and after the reaction was then calculated to obtain the difference map and used for subsequent analysis. The CSA analysis was repeated 12 times for each sample.
(1)$$\Delta R=\left|dR_{a}-dR_{b}\right|$$
(2)$$\Delta G=\left|dG_{a}-dG_{b}\right|$$
(3)$$\Delta B=\left|dB_{a}-dG_{b}\right|$$
where *a* means after exposure and *b* mean before exposure. *ΔR*, *ΔG*, and *ΔB* are the color differences.

### 2.9. Statistical Analysis

The results were presented as the mean and standard deviation of each experiment. The principal component analysis (PCA) and linear discriminant analysis (LDA) were used to classify pickle samples with different fermentation degrees to visualize the data results obtained from the sensor array. In addition, the partial least squares regression (PLSR) was used to determine the correlation between pH, total acid, lactic acid, volatile acids, and CSA data. Charts and PCA were generated using Origin Software (Origin Lab Corporation, Northampton, MA, USA). LDA was performed using SPSS (SPSS Inc., Chicago, IL, USA). PLSR was performed using the Unscrambler X (Camo Corp., Oslo, Norway).

## 3. Results and Discussion

### 3.1. Screening of Starter Cultures

At low salt concentrations, all strains showed good growth ability and acid production (Figure 2a,b). Among them, the acid production ability of LZ-395, LP-165, LP-046, and LP-567 was significantly higher than that of other strains, and the growth ability of LP-046 was the strongest. At a high salt concentration, the vitality of all strains was inhibited, while the acidification capacity was progressively reduced. Only LZ-395, LP-165, LP-046, and LP-567 were still able to grow at 8% salt concentrations.

When vegetable juice was used as a medium, LZ-395 and LP-165 displayed excellent acid production capacity at 12 h, and the acidity of LP-046, and LP-567 rose with the extension of time to 48 h. At this point, the acidification capacity from strong to weak was LP-046, LP-165, LP-567, LZ-395, LM-216, LS-620, and LB-373. There was no considerable fluctuation in their concentration of bacteria (Figure 2c,d).

After comprehensive analysis of the above data, LP-165 was eventually selected as the starter culture for subsequent fermentations because of its excellent acidification capacity and salt tolerance.

### 3.2. Biochemical Change during Fermentation

The pH, an important indicator that reflects the progress of fermentation, decreased with the extension of time. As shown in Figure 3a, the pH value decreased significantly (*p* < 0.05) from 5.56 to 4.01 in the first five days of fermentation, and then was kept around 4.0. Total acid was negatively correlated with pH value. Initially, the total acid was around 0.31 g/100 g, then increased significantly (*p* < 0.05) to 0.61 g/100 g on the second day, then increased significantly (*p* < 0.05) to 0.91 g/100 g on the fifth day, with no significant change (*p* > 0.05) after that. The nitrite content dropped rapidly (*p* < 0.05) from 17.28 to 3.29 mg/kg in the first three days, and then increased slightly to maintain a level of about 2 mg/kg (Figure 3b). During fermentation, the nitrite content of pickles was significantly lower than the safety level of 20 mg/kg stipulated in the Chinese national standard (GB 2762-2017). The salinity was 5.36 g/100 g at the beginning of fermentation, increasing significantly (*p* < 0.05) to 6.63 g/100 g on the fifth day, and then dropping slightly (Figure 3c). The LAB concentration was 7.71 Lg (CFU/g) at the beginning of the fermentation, and it accumulated to 8.20 Lg (CFU/g) on the first day (Figure 3d). The elevation of LAB leads to the accumulation of lactic acid, which in turn inhibits LAB. There was a significant (*p* < 0.05) decrease after the third day. Finally, the LAB number was reduced to 5.67 Lg (CFU/g).

The values of pH and total acid of the Apo pickle changed in a very regular manner consistent with the studies of Wang et al. [27]. The acidity showed high growth from the beginning, probably because *L. Plantarum* was added as a starter culture, which has the characteristic of fast acid production [23,28]. This suggests that it is feasible to end fermentation quickly by adding starter culture. Artificial inoculation led to a rapid increase in the number of *L. Plantarum*, resulting in a large accumulation of lactic acid, which in turn inhibited the growth of lactic acid bacteria [29]. Nitrite is a major hazard in pickled vegetables, which is mainly produced by nitrate-reducing bacteria. The drastic decrease of nitrite levels was mainly due to the increased of lactic acid bacteria, which degraded nitrite [30,31,32]. In the early fermentation, salt penetrated the vegetable tissues, and the salt concentration increased. In the middle and late stages, this showed a relatively stable trend, but lower than 7% [33]. Fermentation was completed when the pH value fell below 4.0 (total acid exceeds 0.70 g/100 g) [34,35]. At this time, other indicators of the Apo pickle also indicated that it had finished fermentation.

### 3.3. Change of Organic Acid during Fermentation

The normal microbial fermentation is mainly lactic acid, accompanied by a small amount of alcohol fermentation and a small amount of acetic acid fermentation. Organic acids are mainly lactic acid and acetic acid [36], and the contents of lactic acid and acetic acid are similar on the first day [24]. After 1 day of fermentation, the acetic acid content increased rapidly (*p* < 0.05) by 3.78 mg/mL and then started to decrease and maintain a level of approximately 3.0 mg/mL. This is because the acetic acid in the fermentation process reacts with alcohol to produce esters that improve the flavor of pickles [37]. Lactic acid increased (*p* < 0.05) to 3.64 mg/mL after 2 days of fermentation and then slowly increased (*p* > 0.05) to 4.17 mg/mL after 7 days of fermentation, after which it decreased. This was consistent with the changing trend of total acid. The content of succinate was very low and did not fluctuate. Citric acid was reduced to 0 mg/mL after 5 days of fermentation (Table 3).

These organic acids affect the flavor of pickles. The lactic acid in organic acids is originally a good flavoring agent with a mild and slightly sour taste [38]. Although acetic acid has stimulation, the appropriate amount of acetic acid can play the role of seasoning. Citric acid has a refreshing flavor, but the content is low. It can be speculated that the acidity of inoculated fermented products is sharper and less refreshing.

### 3.4. Analysis of Volatile Organic Compounds by the GC–MS

A total of 192 VOCs from eleven classes were measured during pickle fermentation, including 22 alcohols, 28 esters, 10 acids, 12 sulfur compounds, 47 aldehydes and ketones, 8 nitriles, and other compounds (Figure 4).

Flavor substances of pickles partly come from the raw materials, and partly from the fermentation products. The Apo pickle were made from *Brassica* spp., in which isothiocyanates, thiocyanates, and nitrites are major sources of pungent aroma and bitter taste [39]. At the beginning of fermentation, the flavor substance of the Apo pickle mainly came from the raw material itself [40], including isothiocyanates and thiocyanates, and the high concentration was phenethyl isothiocyanate, accounting for 27.05%. With the increase of fermentation time, thiocyanate further decomposed to form small molecules and the content decreased rapidly (*p* < 0.05), being only 3.04% on the first day. Esters are derived from the reaction of alcohols with carboxylic acids, which help produce pleasant, fruity, and floral odors at lower thresholds [41]; these were mainly methyl palmitate, ethyl palmitate, and phenethyl acetate, among others. Although many VOCs were detected, there were not many significant changes with fermentation, which may be related to the single fermentation system [42]. In addition, the most variable part of the fermentation process is the volatile acid. The content and type of acid gradually increases with the fermentation time. The percentage of acetic acid was only 0.67% before fermentation, significantly rising (*p* < 0.05) to 9.93% the first day and to 27.72% on the fifteenth day. The type of acids gradually increased with the fermentation time [33,43]. Five acids were tested on the fifteenth day, including acetic acid, capric acid, hexanoic acid, nonanoic acid, and octanoic acid. However, the largest proportion was still acetic acid (12.97%). The high level of acetic acid formed during fermentation was a representative factor affecting the sourness of pickles, which was formed by *L. Plantarum* [44]. According to the biochemical changes it was considered that by five days’ fermentation, the Apo pickle had reached the maturity standard, and the largest proportion of VOCs thereafter were acetic acid. A similar conclusion was found in the report of Choi et al. [24].

### 3.5. Analysis of Volatile Organic Compounds by the CSA

Unlike GS–MS, which can recognize volatile gas specificity, CSA is a nonspecific reaction, and cross-reaction with VOCs of the sample can occur. This is caused by the principle of the colorimetric array, in which each dye reacts with a variety of gases, and each gas reacts with multiple dyes. Pickles contain a variety of flavor compounds, and different fermentation states have different VOCs. Using this artificial olfactory technique based on the CSA, we realized olfactory visualization [45]. Figure 5 shows the difference images for the 0th, 1st, 2nd, 3rd, 5th, 7th, 11th, and 15th days. It could be seen that the fingerprint of varying fermentation time is different, so it is feasible to distinguish fermentation states by the CSA.

#### 3.5.1. Classification Result of the PCA

PCA is a common technique for reducing the dimensionality of numerical datasets, and can transform a large number of related variables into a few unrelated variables, namely, the principal component (PC). For PCA of the CSA results (Figure 6b), it can be seen that the results on day 0 are compact and have no intersection with other samples. However, the overall trend shows that on PC2, as the fermentation progresses, the sample points will diffuse to the outside, and the early stage will be around the center. In general, the classification results are not ideal, which is related to rapid fermentation.

#### 3.5.2. Classification Result of the LDA

Figure 6c is a two-dimensional plot of LD1 and LD2; all sample clusters were visualized in this plot. LDA divided the samples at different times into four groups: 0 d, 1 d, 2–3 d, and 5–15 d. The points within the group were clustered, while there was an obvious clustering trend among different groups. The pickle samples with different fermentation times were successfully distinguished by LDA, and each group was separated. This classification result is similar to that of pH classification, but because the VOCs of 0 d have a unique flavor, they are not divided into a group with 1 d. The accuracy of cross-validation classification reached 92.7% (Table 4), which proved that the CSA method could distinguish the different stages of Apo pickle fermentation.

The LDA and the PCA algorithms are both common dimensionality reduction techniques, but there are differences between them. First, PCA is unsupervised and selects the direction with the largest variance of data after projection. Therefore, the larger variance the PCA explained, the more information it represents. We can achieve the purpose of dimensionality reduction and remove redundant dimensions by using PCs to represent the original data. LDA is supervised to select the direction with small intro-class variance and large inter-class variance after projection. To find the discriminant dimensions in the data, different categories can be distinguished as much as possible after the original data is projected in these directions [46,47,48]. In this study, the first two PCs explained 42.65% of the total variance and the first 10 PCs explained 84.85% (Figure 6a), indicating that PCA could not fully reflect the information in the original data. In contrast, LDA could explain 64.2% (Table 5) of the total variance. PCA classification results were not ideal, which might be because most indicators in the CSA reacted with VOCs in pickles, and each indicator had a unique image of VOCs. However, there were few collinear variables in the colorimetric sensor data. If the redundant information in the original data is insufficient, then it is not obvious to reduce the dimension of the data by using PCA. Xu et al. have also reported that PCA is not effective in classifying the degree of yeast fermentation, because fermentation is a complex process that also exhibits the same VOCs at different degrees [29]. However, LDA takes into account and exploits differences in population means and tends to outperform PCA in terms of population separation [46]. Therefore, LDA achieved better results than PCA.

#### 3.5.3. PLSR Analysis

The results (Figure 6d) showed that LDA could be used to successfully distinguish the different degrees of fermentation of Apo pickles. In addition, PLSR was an effective tool for determining whether the CSA could predict the fermentation acids of pickles. In the PLSR model, the coefficient of determination and root mean square error (RMSE) were related to accuracy and precision, respectively [49]. The pH, total acids, lactic acid, and concentration of volatile acids were successfully predicted by the CSA. The correlation coefficients of the calibration sets (Rc^2^) were 0.820, 0.761, 0.796, and 0.815. The correlation coefficients of the prediction sets (Rp^2^) were 0.724, 0.642, 0.662, and 0.719. The RMSEs of the calibration sets were 0.229, 0.111, 0.307, and 3.365. The RMSEs of the prediction sets were 0.290, 0.137, 0.396, and 4.237. The results demonstrated that the CSA had a good correlation with acids, showing good predictive ability.

## 4. Conclusions

The results showed that the use of *L. Plantarum* was able to rapidly reach the ending of the fermentation of the Apo pickle in five days, and it can be used in factories. CSA is very suitable for the detection of pickles at different stages, the accuracy of backward substitution verification can reach 99%. Therefore, CSA could be used to monitor the fermentation of Apo pickles. In future research, CSA will be further improved and made into an intelligent label placed in the Apo fermentation container to judge the fermentation stage through real-time image acquisition.

## Figures and Tables

**Figure 1 foods-11-03577-f001:**
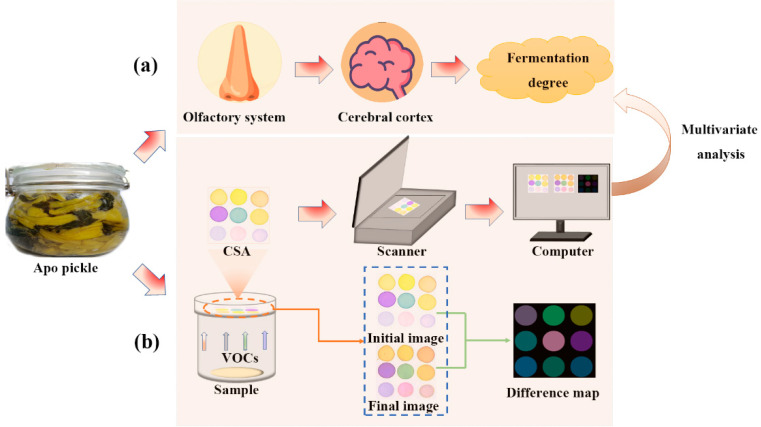
Schematic diagram of the colorimetric sensors array (CSA). (**a**) Determination of fermentation degree by the olfactory system; (**b**) determination of fermentation status by using CSA.

**Figure 2 foods-11-03577-f002:**
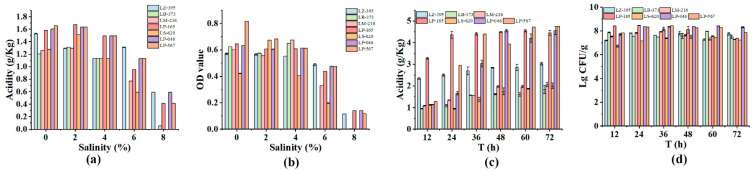
Screening of starter cultures. (**a**,**b**) Acidity and optical density (OD) values of the strain at different salt concentrations; (**c**,**d**) acidity and Lactobacillus number of strains at different time.

**Figure 3 foods-11-03577-f003:**
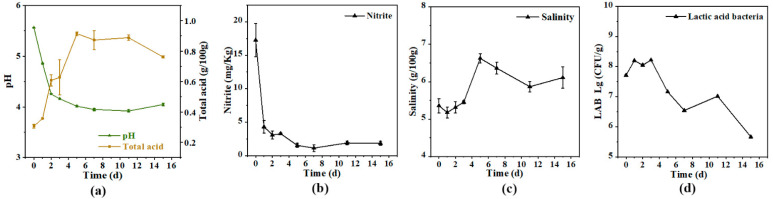
Biochemical of Apo pickle during fermentation. (**a**) pH and total acids; (**b**) nitrite; (**c**) salinity; (**d**) lactic acid bacteria.

**Figure 4 foods-11-03577-f004:**
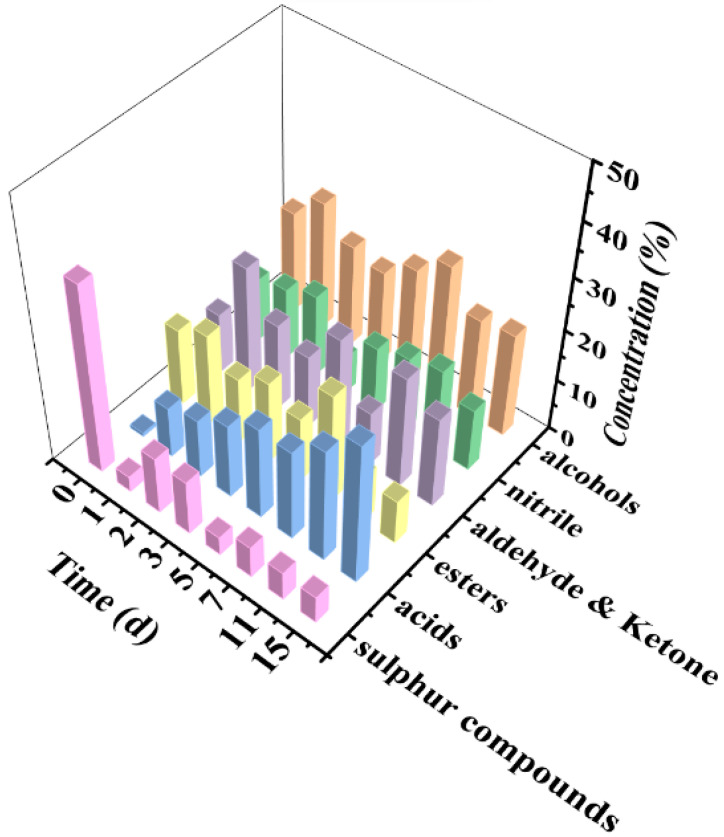
Category distributions of volatile organic compounds of pickle during fermentation.

**Figure 5 foods-11-03577-f005:**
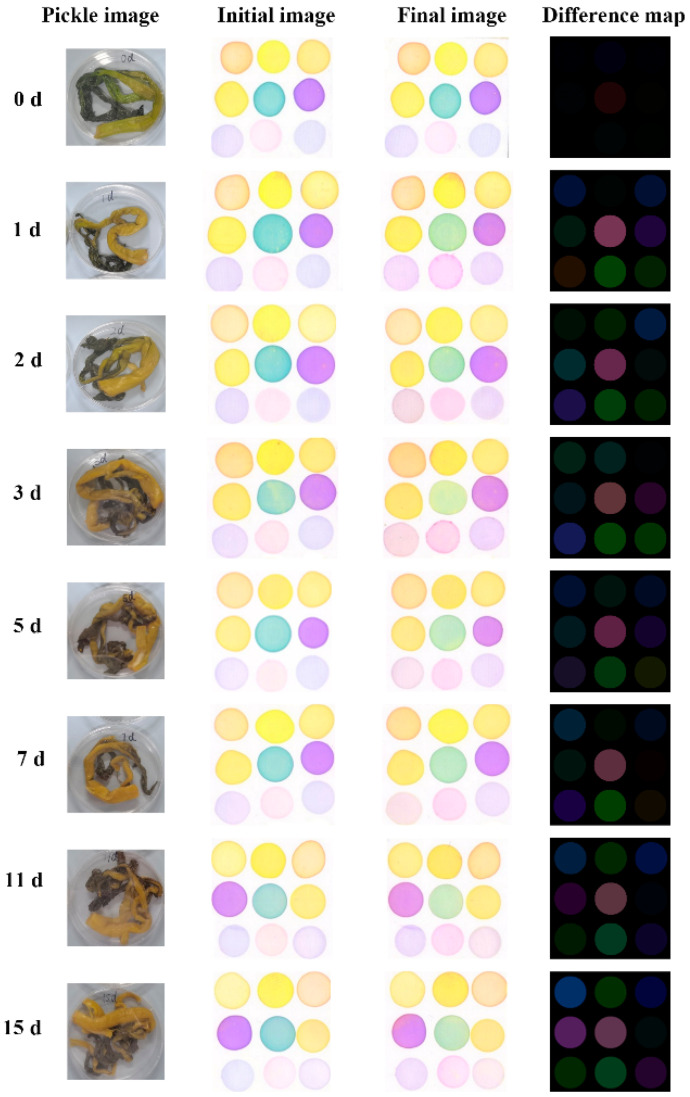
Images of CAS for pickles at different fermentation times.

**Figure 6 foods-11-03577-f006:**
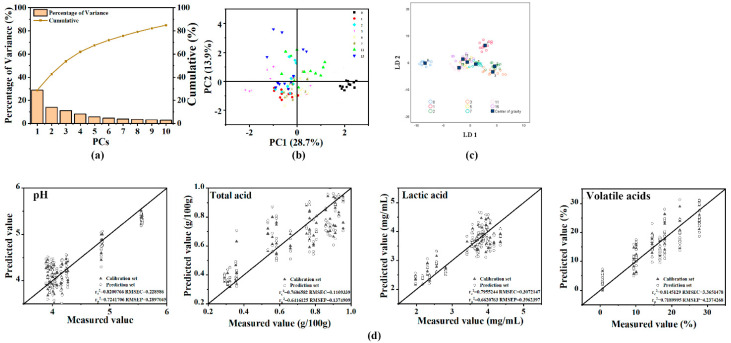
Data analysis of Apo pickle by CAS detection. (**a**) Rate of contribution the first 10 principal components (PCs) of the total variance; (**b**) data grouped by days were analyzed by principal component analysis (PCA); (**c**) data grouped by days were analyzed by linear discriminant analysis (LDA); (**d**) the partial least squares regression (PLSR) prediction result plot of pH, total acids, lactic acid and volatile acids.

**Table 1 foods-11-03577-t001:** The seven strains used in the experiment.

Strain Code	Species
LP-046	*Lactobacillus plantarum*
LP-567	*Lactobacillus plantarum*
LP-165	*Lactobacillus plantarum*
LS-620	*Lactobacillus sakei*
LM-216	*Leuconostoc mesenteroides*
LB-373	*Lactobacillus brevis*
LZ-395	*Levilactobacillus zymae*

**Table 2 foods-11-03577-t002:** Nine kinds of gas-sensitive materials.

Number	Gas-Sensitive Material
1	Bromocresol green ^b^
2	Methyl Red sodium salt ^b^
3	Bromophenol blue ^b^
4	Chlorphenol red ^b^
5	Bromothymol blue sodium ^b^
6	Bromcresol purple ^a^
7	Bromophenol blue ^c^
8	Chlorphenol red ^c^
9	Bromcresol purple ^c^

^a^: Untreated. ^b^: pH adjustment. ^c^: sulfur derivatization.

**Table 3 foods-11-03577-t003:** Changes of organic acids in fermentation.

Time (d)	Lactic Acid (mg/mL)	Acetic Acid (mg/mL)	Citric Acid (mg/mL)	Succinic Acid (mg/mL)	Fumaric Acid (mg/mL)
0	2.17 ± 0.20 ^f^	2.26 ± 0.31 ^d^	0.43 ± 0.06 ^b^	0.06 ± 0.02 ^ab^	0.01 ± 0.00 ^a^
1	2.56 ± 0.26 ^d^	3.78 ± 0.04 ^a^	0.55 ± 0.03 ^a^	0.05 ± 0.01 ^b^	0.01 ± 0.00 ^a^
2	3.64 ± 0.07 ^c^	3.08 ± 0.11 ^b^	0.53 ± 0.01 ^a^	0.07 ± 0.00 ^a^	-
3	3.87 ± 0.00 ^bc^	3.30 ± 0.10 ^bc^	0.57 ± 0.02 ^a^	0.07 ± 0.01 ^ab^	-
5	3.84 ± 0.11 ^bc^	3.20 ± 0.04 ^bc^	-	0.07 ± 0.01 ^a^	-
7	4.17 ± 0.18 ^a^	3.04 ± 0.15 ^bc^	-	0.06 ± 0.00 ^ab^	-
11	4.01 ± 0.05 ^ab^	2.96 ± 0.20 ^c^	-	0.06 ± 0.20 ^ab^	-
15	3.55 ± 0.14 ^c^	2.93 ± 0.07 ^c^	-	0.06 ± 0.00 ^ab^	-

Different letters indicate a significant difference between the values in the same tested items (*p* < 0.05, by least significant difference test). Not detected.

**Table 4 foods-11-03577-t004:** Classification results of CSA data of samples at different fermentation time by LDA.

Time (d)	0	1	2	3	5	7	11	15	Total Accuracy (%)
Backtracking validation (%)	100	100	100	100	91.7	100	100	100	99.0
Cross validation (%)	100	100	83.3	100	75.0	83.3	100	100	92.7

**Table 5 foods-11-03577-t005:** Eigenvalues and variances of LDA.

Function	Eigenvalues	Percentage of Variance (%)	Cumulative Percentage (%)
1	16.992	43.3	43.3
2	8.211	20.9	64.2
3	6.197	15.8	80.0
4	3.969	10.1	90.1
5	1.638	4.2	94.3
6	1.316	3.4	97.7
7	0.919	2.3	100.0

## Data Availability

Data is contained within the article.

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
