# Peer review of "Detection of the Inoculated Fermentation Process of Apo Pickle Based on a Colorimetric Sensor Array Method"

_foods, 2022, doi:10.3390/foods11223577_

Round 1

Reviewer 1 Report

The manuscript deals with designing a calorimetric sensor array for the detection of the fermentation process. The research methodology, as well as the presentation of the research, is good. The importance of research is well justified in the introduction. However, the manuscript has several errors that need to be corrected. I recommend the manuscript for publication after addressing the following comments.

Abstract

Line 2-3: “There is currently very little information available about the Apo pickle's preparation and testing” write in one sentence what work has been done and the knowledge gap. Only writing “little information“ is absurd and seems casual.

Introduction

Line 10: write – popular in or within.

Line 9-10: write reference.

Line 10-12: Write some examples of application of CSA system in food industry. Better to correlate with present topic.

Authors must give a schematic diagram of colorimetric sensor array system and explain its working principle. Arrangement of component is not clear from the given diagram.

Materials and Methods

Summarize the set of different experimental treatments in a table.

Results and Discussion

Results are presented well and discussed with justification. Authors need to include similar observations or trends by other researchers for other types of pickles. Follow this for all the parameters. This will be helpful in understanding whether the trend is similar or there is any deviation.

Conclusion: Instead of the conclusion of the research, a summery of the research has been written here. Authors should write about the impact of this research. How new processing technique is advantageous over traditional methods and applications of this research findings.

Figure 3: Why not start the value on the Y axis from the origin i.e. “0”?

Figure 6: “PH” should be “pH”; provide legends for abbreviations.

Table 3: Write caption – what is a, b, ab, .. ?

Author Response

Comments and Suggestions for Authors

The manuscript deals with designing a colorimetric sensor array for the detection of the fermentation process. The research methodology, as well as the presentation of the research, is good. The importance of research is well justified in the introduction. However, the manuscript has several errors that need to be corrected. I recommend the manuscript for publication after addressing the following comments.

Response: The authors thank the reviewer for the encouraging comments on the importance of the work. To further provide a better clarity on the concept and to address the reviewer’s comments, we have revised the whole manuscript carefully. The responses to all reviewer comments have been answered point-by-point.

Point 1: Line 2-3: “There is currently very little information available about the Apo pickle's preparation and testing” write in one sentence what work has been done and the knowledge gap. Only writing “little information” is absurd and seems casual.

Response: We agree with the comment and re-wrote the sentence in the revised manuscript as the following:

Apo pickle is a traditional Chinese fermented vegetable. However, the traditional fermentation process of Apo pickle is slow, easy to deteriorate, and the fermentation process cannot be judged in time.

Point 2: Line 10: write – popular in or within.

Response: We apologize for the language problems in the original manuscript. Already modified.

Point 3: Line 9-10: write reference.

Response: We deeply appreciate the reviewer’s suggestion. We have added relevant reference.

Reference

Huang, L.; Wang, M.; Liu, H. Identification of Adulterated Extra Virgin Olive Oil by Colorimetric Sensor Array. Food Analytical Methods 2021, 15, 647-657, doi:10.1007/s12161-021-02141-x.

Kaeppler, K. Crossmodal Associations Between Olfaction and Vision: Color and Shape Visualizations of Odors. Chemosensory Perception 2018, 11, 95-111, doi:10.1007/s12078-018-9245-y.

Point 4: Line 10-12: Write some examples of application of CSA system in food industry. Better to correlate with present topic.

Response: Thank you for your question. We have rewritten and added new references.

Reference

Li, H.; Zhang, B.; Hu, W.; Liu, Y.; Dong, C.; Chen, Q. Monitoring black tea fermentation using a colorimetric sensor array-based artificial olfaction system. Journal of Food Processing and Preservation 2018, 42, doi:10.1111/jfpp.13348.

Chen, Q.; Liu, A.; Zhao, J.; Ouyang, Q.; Sun, Z.; Huang, L. Monitoring vinegar acetic fermentation using a colorimetric sensor array. Sensors and Actuators B: Chemical 2013, 183, 608-616, doi:10.1016/j.snb.2013.04.033.

Dai, C.; Huang, X.; Huang, D.; Lv, R.; Sun, J.; Zhang, Z.; Aheto, J.H. Real‐time detection of saponin content during the fermentation process of Tremella aurantialba using a homemade artificial olfaction system. Journal of Food Process Engineering 2019, 42, doi:10.1111/jfpe.13101.

Point 5: Authors must give a schematic diagram of colorimetric sensor array system and explain its working principle. Arrangement of component is not clear from the given diagram.

Response: Thank you for your comment. This has been clarified in the revised version of the manuscript, Figure 1 has been redrawn for inclusion in the revised manuscript.

Figure 1. Schematic diagram of the colorimetric sensors array (CSA). (a), Determination of fermentation degree by the olfactory system; (b), Determination of fermentation status by using CSA.

Point 6: Summarize the set of different experimental treatments in a table.

Response: Thank you for your comment, we have rewritten the experimental treatments (page 8) and made corresponding changes in Table 2.

Table 2. Nine kinds of gas-sensitive materials

Number

Gas-sensitive material

1

Bromocresol green b

2

Methyl Red sodium salt b

3

Bromophenol blue b

4

Chlorphenol red b

5

Bromothymol blue sodium b

6

Bromcresol purple a

7

Bromophenol blue c

8

Chlorphenol red c

9

Bromcresol purple c

a: Untreated

b: pH adjustment

c: sulfur derivatization

Point 7: Results are presented well and discussed with justification. Authors need to include similar observations or trends by other researchers for other types of pickles. Follow this for all the parameters. This will be helpful in understanding whether the trend is similar or there is any deviation.

Response: Thank you for your suggestion. We have revised the suggested content to the manuscript on Results and Discussion (Pages 12-21).

Point 8: Instead of the conclusion of the research, a summary of the research has been written here. Authors should write about the impact of this research. How new processing technique is advantageous over traditional methods and applications of this research findings.

Response: Thank you for your suggestion. We have revised the conclusion of this paper.

The results showed that the use of L. Plantarum was able to rapidly reach the ending of the fermentation of the Apo pickle in five days and can be used in factories. CSA is very suitable for the detection of pickles at different stages, the accuracy of backward substitution verification can reach 99%. Therefore, CSA could be used to monitor the fermentation of Apo pickle. In future research, CSA will be further improved and made into an intelligent label placed in the Apo fermentation container to judge the fermentation stage through real-time image acquisition.

Point 9: Figure 3: Why not start the value on the Y axis from the origin i.e. “0”?

Response: One is that this can more clearly see the change, and the second is that for aesthetic purposes, the data is far from the zero point and there will be a lot of white space.

Point 10: Figure 6: “PH” should be “pH”; provide legends for abbreviations.

Response: Thank you for your comment. We have made the corresponding changes in Figure 6.

Point 11: Table 3: Write caption – what is a, b, ab, .. ?

Response: Thank you very much for your comment. Different letters indicate a significant difference between the values in the same tested items (p < 0.05, by least significant difference test). And this explanation is attached as a footnote in Table 3.

Reviewer 2 Report

The manuscript is written with clear understanding of the project addressed. However, there are some concerns that need to be addressed to enhance the quality of the manuscript. My specific comments are as follows:

Abstract:

Elaborate more on methods of your study.

Introduction:

“Unlike Sichuan pickle, which is fermented in brine at a low salt concentration (20–100 g/L) by the…” add citation

“Pickles are usually fermented spontaneously by lactic acid bacteria (LAB), acetic acid bacteria, yeast, and other environmental microorganisms.” Add citation

Add literatures on the application of CSA

Add main objective at the end of introduction

Based on your objectives, please compare how your study is different from those that have already been published

Materials and methods:

How many cabbage samples were used?

Results and discussion:

L14-21: Delete. This is not discussion based on your results. Please revise

Elaborate on the fluctuation of biochemical change during fermentation

Similar with section 3.3. elaborate on the fluctuation and significant difference

“Using this artificial olfactory echnique based on the CSA, we realized olfactory visualization.” Explain

The findings lack in terms of justification and major findings in all subsection in Results & discussion.

Conclusions:

Highlight on the main finding of your study

General comments:

Please check the reference styles and grammar of the manuscript.

Please follow the journal’s guideline for format

Author Response

Comments and Suggestions for Authors: The manuscript is written with clear understanding of the project addressed. However, there are some concerns that need to be addressed to enhance the quality of the manuscript.

Response: Our deepest gratitude goes to you for your careful work and thoughtful suggestions that have helped improve this paper substantially. We have carefully revised the language issue and revised the manuscript to address the review comments. The responses to all reviewer comments have been answered point-by-point.

Point 1: Elaborate more on methods of your study.

Response: Thanks for your suggestion. We revised the abstract, try to introduce more information within the limitation of 200 words.

Apo pickle is a traditional Chinese fermented vegetable. However, the traditional fermentation process of Apo pickle is slow, easy to deteriorate, and the fermentation process cannot be judged in time.  To improve fermentation, the LP–165 (L. Plantarum) with high salt tolerance, acidification, and growth capacity was chosen as the starter culture. Meanwhile, A colorimetric sensor array (CSA) sensitive to pickle volatile compounds was developed to differentiate Apo pickles with varying degrees of fermentation. The color components were extracted from each dye in color change profiles and were analyzed using principal component analysis (PCA) and linear discriminant analysis (LDA). The fermentation process of Apo pickle was classified into four phases by LDA. The accuracy of backward substitution verification was 99% and the accuracy of cross-validation was 92.7%. Furthermore, the partial–least–squares regression (PLSR) showed data from CSA were correlated with pH total acid, lactic acid, volatile acids of Apo pickle. These results illustrated that the CSA reacted quickly to inoculated Apo pickle and could be used to detect fermentation.

Point 2: “Unlike Sichuan pickle, which is fermented in brine at a low salt concentration (20–100 g/L) by the…” add citation.

Response: We are grateful for the suggestion. We have added a new reference.

Reference

Xiong, T.; Guan, Q.; Song, S.; Hao, M.; Xie, M. Dynamic changes of lactic acid bacteria flora during Chinese sauerkraut fermentation. Food Control 2012, 26, 178-181, doi:10.1016/j.foodcont.2012.01.027.

Point 3: “Pickles are usually fermented spontaneously by lactic acid bacteria (LAB), acetic acid bacteria, yeast, and other environmental microorganisms.” Add citation.

Response: Thank you for your comments. We have added a new reference.

Reference

Lavefve, L.; Marasini, D.; Carbonero, F. Microbial Ecology of Fermented Vegetables and Non-Alcoholic Drinks and Current Knowledge on Their Impact on Human Health. Adv Food Nutr Res 2019, 87, 147-185, doi:10.1016/bs.afnr.2018.09.001.

Point 4: Add literatures on the application of CSA.

Response: Thank you very much for your question. We have added more literatures about CSA applications.

Reference

Li, H.; Zhang, B.; Hu, W.; Liu, Y.; Dong, C.; Chen, Q. Monitoring black tea fermentation using a colorimetric sensor array-based artificial olfaction system. Journal of Food Processing and Preservation 2018, 42, doi:10.1111/jfpp.13348.

Chen, Q.; Liu, A.; Zhao, J.; Ouyang, Q.; Sun, Z.; Huang, L. Monitoring vinegar acetic fermentation using a colorimetric sensor array. Sensors and Actuators B: Chemical 2013, 183, 608-616, doi:10.1016/j.snb.2013.04.033.

Dai, C.; Huang, X.; Huang, D.; Lv, R.; Sun, J.; Zhang, Z.; Aheto, J.H. Real‐time detection of saponin content during the fermentation process of Tremella aurantialba using a homemade artificial olfaction system. Journal of Food Process Engineering 2019, 42, doi:10.1111/jfpe.13101.

Point 5: Add main objective at the end of introduction.

Response: We deeply gratitude to you for your suggestions. In accordance with the comments, we have revised the last paragraph of the introduction by adding the main objectives.

The aim of this study is to screen a strain to improve the fermentation of Apo pickle and prepare a colorimetric sensor array (CSA) to distinguish the fermentation process in a visual and non-destructive method.

Point 6: Based on your objectives, please compare how your study is different from those that have already been published.

Response: Thank you for your comment, we have added this section.

The advantage of this method is that it is real time, fast response, does not require pre-treatment of the sample and can effectively determine the condition of the food. At present, CSA has been successfully applied to tea [20], vinegar [21], and tremella aurantialba [22]. Based on the characteristics of CSA, it can also be used to detect the fermentation process of the Apo pickle.

Point 7: How many cabbage samples were used?

Response: A total of 10 Kg of cabbage was used, and the fermentation was divided into 100 mL fermentation jars (130 g).

Point 8: L14-21: Delete. This is not discussion based on your results. Please revise.

Response: Thank you for your valuable question. We have revised.

Point 9: Elaborate on the fluctuation of biochemical change during fermentation.

Response: Thank you very much for your suggestion. We have revised section 3.2 on pages 12-13.

Point 10:  Similar with section 3.3. elaborate on the fluctuation and significant difference.

Response: Thank you very much for your suggestion. We have revised section 3.4 on pages 15-17.

Point 11: “Using this artificial olfactory echnique based on the CSA, we realized olfactory visualization.” Explain.

Response: Thank you for your valuable question. The CSA simulates the human olfactory system, presenting different combined responses to different compounds based on cross-reactive sensing receptors. While CSA presents the results visually by color change, it is also called this technology as olfactory visualization technology.

Reference

  1. Huang, L.; Wang, M.; Liu, H. Identification of Adulterated Extra Virgin Olive Oil by Colorimetric Sensor Array. Food Analytical Methods 2021, 15, 647-657, doi:10.1007/s12161-021-02141-x.
  2. Kaeppler, K. Crossmodal Associations Between Olfaction and Vision: Color and Shape Visualizations of Odors. Chemosensory Perception 2018, 11, 95-111, doi:10.1007/s12078-018-9245-y.

Point 12: The findings lack in terms of justification and major findings in all subsection in Results & discussion.

Response: Thank you for your suggestion. We have added the suggested content to the manuscript on pages 11-17.

Point 13: Highlight on the main finding of your study.

Response: This suggestion is appreciated. We have revised the conclusion section.

The results showed that the use of L. Plantarum was able to rapidly reach the ending of the fermentation of the Apo pickle in five days and can be used in factories. CSA is very suitable for the detection of pickles at different stages, the accuracy of backward substitution verification can reach 99%. Therefore, CSA could be used to monitor the fermentation of Apo pickle. In future research, CSA will be further improved and made into an intelligent label placed in the Apo fermentation container to judge the fermentation stage through real-time image acquisition.

Point 14: Please check the reference styles and grammar of the manuscript.

Response: Thank you for your suggestion. We have adopted the MDPI reference format and carefully corrected grammar issues.

Point 15: Please follow the journal’s guideline for format.

Response: Thank you for your comments. We have modified the article format according to the author guidelines.

Reviewer 3 Report

The manuscript develops a colorimetric sensor array (CSA) sensitive to pickle volatile compounds, but there are some deficiencies for the CSA now. The CSA only detects the le volatile compounds but do not overall assess the indicators of pickle, such as nitrite. Therefore, in my opinion, there are some more work to do before the CSA could be used to detect fermentation of Apo pickle.

Author Response

Comments and Suggestions for Authors: The manuscript develops a colorimetric sensor array (CSA) sensitive to pickle volatile compounds, but there are some deficiencies for the CSA now. The CSA only detects the le volatile compounds but do not overall assess the indicators of pickle, such as nitrite. Therefore, in my opinion, there are some more work to do before the CSA could be used to detect fermentation of Apo pickle.

Response: Thanks for your question. The typical fermentation duration of Apo pickle is lengthy, and little attention has been made to its fermentation degree. As a result, one of the goals of our study is to speed fermentation by inoculation. In terms of pickle testing, most individuals concentrate on taste and microbiological as the primary research. We sought to discover a technique that could be used in fermentation assays that is quick and does not harm the material. Our trials demonstrated that this strategy is practical. Furthermore, because Apo pickle is dry fermented, there is no brine, thus testing for non-volatile chemicals necessitates pre-treatment, which may have harmed the sample. Volatile and non-volatile compounds are also connected with lactic acid bacteria during the fermentation process. At the conclusion of fermentation, acid generation by lactic acid bacteria increases overall acid, which inhibits dangerous bacteria and reduces the level of some harmful compounds (such as nitrite). So, the detection of volatile gases can yield a lot of information of fermentation. In addition, studies using CSA to determine the fermentation status of pickles are very limited. And we currently have gas-liquid tests for pickled radish and will further explore the application of CSA to foods in the future.

Round 2

Reviewer 1 Report

I read the revised manuscript thoroughly. I appreciate the efforts made by the authors in improving the manuscript.

I find that most of the suggestions are addressed and I am satisfied with the correction made.

I recommend the manuscript for publication.

Reviewer 2 Report

The authors have addressed the comments. Hence, the paper can be accepted.

Reviewer 3 Report

Accept in present form